



# Estimating mean molecular weight, carbon number, and OM/OC with mid-infrared spectroscopy in organic particulate matter samples from a monitoring network

Amir Yazdani[1], Ann M. Dillner[2], and Satoshi Takahama[1]

[1]ENAC/IIE Swiss Federal Institute of Technology Lausanne (EPFL), Lausanne, Switzerland
[2]Air Quality Research Center, University of California Davis, Davis, California, USA

**Correspondence:** Satoshi Takahama (satoshi.takahama@epfl.ch)

**Abstract.** Organic matter (OM) is a major constituent of fine particulate matter which contributes significantly to degradation of visibility, radiative forcing, and causes adverse health effects. However, due to its sheer compositional complexity, OM is difficult to characterize in its entirety. Mid-infrared spectroscopy has previously proven useful in the study of OM by providing extensive information about functional group composition with high mass recovery. Herein, we introduce a new method for ob-

taining additional characteristics such as mean carbon number and molecular weight of these complex organic mixtures using the aliphatic $C-H$ absorbance profile in mid-infrared spectrum. We apply this technique to spectra acquired non-destructively from Teflon filters used for fine particulate matter quantification at selected sites of Inter-agency Monitoring of PROtected Visual Environments (IMPROVE) network. Since carbon number and molecular weight are important characteristics used by recent models to describe evolution in OM composition, this technique can provide semi-quantitative, observational constraints of these variables at the scale of the network. For this task, multivariate statistical models are trained on calibration spectra pre-

pared from atmospherically relevant laboratory standards and are applied to ambient samples. Then, the physical basis linking the absorbance profile of this relatively narrow region in the mid-infrared spectrum to the molecular structure is investigated using a classification approach. The multivariate statistical models predict mean carbon number and molecular weight that are consistent with previous values of organic-mass-to-organic-carbon (OM/OC) ratios estimated for the network using different approaches. The results are also consistent with temporal and spatial variations in these quantities associated with aging pro-

cesses, and different source classes (anthropogenic, biogenic, and burning sources). For instance, the models estimate higher mean carbon number for urban samples and smaller, more fragmented molecules for samples in which substantial aging is anticipated.

## 1 Introduction

### 1.1 Organic aerosols and measurement methods

Organic mass is known to be an important constituent of fine particulate matter (PM). It is estimated to constitute 20–50 % of the total fine PM at mid-latitudes and up to 90 % in tropical forests (Kanakidou et al., 2005). This organic fraction



contributes significantly to aerosol-related phenomena such as visibility and climate change, through radiative forcing and affecting cloud formation, and causes adverse health effects (Shiraiwa et al., 2017b; Hallquist et al., 2009). Such effects underscore the importance of better quantification of organic fraction in particulate matter which is a complex mixture of multitude of compounds whose compositions, concentrations, and formation mechanisms are not yet completely understood
(Turpin et al., 2000).

The determination of organic aerosol composition involves a large range of analytical and computational techniques. Among the widely known techniques are gas chromatography/mass spectrometry (GC/MS), mid-infrared spectroscopy (often referred to as Fourier transform infrared spectroscopy (FT-IR)) and aerosol mass spectrometry (AMS). GC/MS provides molecular speciation information but is limited to a small mass fraction of the organic aerosols as low as 10 % (Hallquist et al., 2009).
AMS and FT-IR, however, can be used to analyze most of the organic mass in addition to providing information about either chemical classes or functional groups (Hallquist et al., 2009). AMS is an on-line technique with a relatively high size and time resolution. Nevertheless, the extensive fragmentation caused by commonly used ionization method in AMS, i.e. electron impact (IE) ionization, makes the identification of original species difficult (Canagaratna et al., 2007; Faber et al., 2017). There are a few emerging soft ionization methods such as electrospray ionization (ESI), and chemical ionization (CI) that minimize
analyte fragmentation but often have other shortcomings such as ionization efficiency, which varies by molecule (Nozière et al., 2015; Iyer et al., 2016; Hermans et al., 2017; Lopez-Hilfiker et al., 2019).

In mid-infrared spectroscopy, the vibrational modes of organic molecules, whose frequencies fall in the range of mid-infrared electromagnetic radiation, are excited. The advantages of mid-infrared spectroscopy over other common techniques of quantifying OM are providing direct information on functional groups, minimizing sample alteration during the analysis, and
having low sampling and analytical cost (Ruthenburg et al., 2014). In previous studies, different statistical methods were used to connect mid-infrared absorbances to molar abundance of different functional groups, from which OM, OC (organic carbon), and the OM/OC ratio were calculated with minimal assumptions (Coury and Dillner, 2008; Ruthenburg et al., 2014; Takahama et al., 2016; Boris et al., 2019). These studies showed good agreement between FT-IR measurements and other methods of OM characterization. For example, Boris et al. (2019) showed that OC measured by FT-IR is around 80 % of OC from thermal
optical reflectance (TOR) measurements.

In addition to the abundance of organic functional groups, mid-infrared spectroscopy is informative about the environment in which organic bonds are vibrating (e.g., degree of hydrogen bonding; Pavia et al. (2008)), therefore can be used to extract more detailed structural information about OM. This ability of mid-infrared spectroscopy has been investigated to a lesser extent in the context of atmospheric OM. In this work, we used this aspect to investigate two important structural parameters in OM, i.e.
mean molecular weight, and mean carbon number. These two parameters are important characteristics used by recent models to describe evolution in atmospheric OM, in terms of its volatility and phase state (Shiraiwa et al., 2017a; Pankow and Barsanti, 2009; Kroll et al., 2011; Donahue et al., 2011). Moreover, inspecting the spatial and temporal variations of these parameters helps us understand the processes involved in aerosol aging, especially fragmentation (Murphy et al., 2012), and can be useful for identification of the dominant sources (Price et al., 2017; Gentner et al., 2012).





In this paper, the mean molecular weight, carbon number, and OM/OC ratio of ambient aerosols, which were collected on polytetrfluoroethylene (PTFE) filters at selected IMPROVE sites, were estimated using FT-IR. First, the aliphatic C−H region (2800–3000 cm$^{-1}$) was extracted from the baseline-corrected spectra of laboratory standards. The C−H spectral bands were then normalized to eliminate abundance information. Then, partial least squares regression (PLSR) was used to develop models

on the high-dimensional and collinear spectral data. Thereafter, the derived models were used to estimate the mean properties of ambient samples. Finally, a classification algorithm was applied to the model estimates to provide a better understanding of how the models function.

## 1.2 Aliphatic C−H absorption and the molecular structure

We have used the aliphatic C−H region (2800–3000 cm$^{-1}$) in mid-infrared spectrum to build models for estimating molecular

weight and carbon number. This section describes the connection of that region of spectrum with the molecular structure of organic aerosols and compares the approach used in this work with previous studies.

Recent studies using FT-IR and AMS have shown that the aliphatic C−H is the most abundant functional group in organic aerosols (Russell et al., 2009; Ruthenburg et al., 2014; Zhang et al., 2007) highlighting its importance in OM. This functional group also exhibits characteristics of "good group" frequencies in mid-infrared stretch region (Mayo et al., 2004). Since the

hydrogen atom is much lighter than the carbon atom, most of the displacement during oscillation is related to the hydrogen; thereby the carbon atom and consequently its connection to the rest of the molecule is involved to a much lesser extent in the stretch (Mayo et al., 2004). This phenomenon results in a fairly consistent profile for C−H absorption band among different molecules containing this functional group and makes it possible to reduce the dimensionality of spectrum to few independent variables describing the band profile (advantageous when constructing models using a limited number of samples). The light

hydrogen atom also causes the aliphatic C−H functional group to absorb at a relatively high stretch frequency, which makes it isolated from most of other absorbing bonds (Mayo et al., 2004) except the broad carboxylic acid O−H stretch that absorbs in the 2400–3400 cm$^{-1}$ range and the ammonium N−H stretch (Pavia et al., 2008). These broad absorption profiles can be separated from the narrow aliphatic C−H bands by baseline correction. The unsaturated and aromatic C−H bonds, which absorb at a slightly higher frequency than aliphatic C−H, were not considered in this work. These bonds are not prevalent in

atmospheric samples (Russell et al., 2011; Decesari et al., 2000) and their absorption usually falls below the FT-IR detection limit (Russell L. M. et al., 2009). The absorption bands attributed to unsaturated and aromatic C−H were not visible in mid-infrared spectra of atmospheric samples of this study.

The aliphatic C−H ($sp^3$-hybridized) stretching band in mid-infrared spectrum is composed of four absorption peaks (two doublets) that are attributed to CH$_2$ (methylene) and CH$_3$ (methyl) symmetric and asymmetric stretches (Mayo et al., 2004).

Methine (tertiary CH) also absorbs in this region, but has a very weak absorption compared to methyl and methylene (Pavia et al., 2008). The profile of these four peaks (characterized by peak frequency, intensity, and width) is affected by the structure of the molecule, inter- and intra-molecular interactions that change electron distribution, and the equilibrium geometry of the molecule (Atkins et al., 2017; Kelley, 2013) as discussed below.



Group vibrational modes in a molecule are not completely decoupled from the rest of the molecule (McHale, 2017). Equation 1 describes a 2-body harmonic oscillator model of molecular vibration (in a classical point of view), for which $\bar{\nu}$ is the fundamental wavenumber at which the bond vibrates, $c$ is the speed of light, $K$ is the spring constant of the chemical bond, $m_H$ is mass of hydrogen atom and $m_M$ is the mass of the rest of the molecule (assuming the rest of the molecule is stiff). The

reduced mass of the system, $\mu$, increases with increasing the molecular weight (Eq. (1)), resulting in a decreased vibrational frequency (wavenumber). There are also effects that change the vibrational frequency through changing the bond strength. For example, electron-withdrawing effect of neighboring polar groups and ring structure strain elevate the absorption frequency of the oscillator by increasing the equivalent spring constant (Pavia et al., 2008). The Bohlmann effect, in which electron density is transferred from the lone pair of a neighboring nitrogen or oxygen into the $C-H$ antibonding orbital, decreases

the frequency by weakening the $C-H$ bond (Lii et al., 2004). Hydrogen-bonding interactions and phase state can also affect absorption frequency and intensity of bands corresponding to vibrational modes (Fornaro et al., 2015; **?**).

$$\bar{\nu} = \frac{1}{2\pi c}\sqrt{\frac{K}{\mu}}, \text{where } \mu = \frac{m_H m_M}{m_H + m_M}. \tag{1}$$

The environment in which the molecules vibrate can effect the absorption peak width through different homogeneous and inhomogeneous broadening mechanisms. Slightly different interaction of molecules in liquids and amorphous solids (to a lesser

extent in crystals) is the basis of inhomogeneous broadening (**?**). This phenomenon determines the change in peak width due to phase state by changing the level of interaction between the molecules. Hydrogen bonding can also cause inhomogeneous broadening due to enhanced anharmonicity (Thomas et al., 2013). The weak hydrogen bond which can exists for aliphatic $C-H$ functional group (Desiraju and Steiner, 2001) broadens its absorption band slightly and shifts its absorption frequency.

The peak height ratios in aliphatic $C-H$ region are also indicators of some structural features of the molecule. For example,

the ratio of peak heights of asymmetric $CH_3$ stretching to asymmetric $CH_2$ stretching shows the relative abundance of these groups in the sample (Orthous-Daunay et al., 2013). For straight-chain alkanes and some polymers, this ratio is directly related to the chain length and can be used to estimate the carbon number of a molecule (Lipp, 1986; Mayo et al., 2004). This ratio as well as the tertiary $C-H$ absorption are informative about the degree of branching in the molecule. The ratio of symmetric to asymmetric $CH_2$ peak heights is an indicator of rotational and conformational order in a molecule, and related to chain

length and phase state (Hähner et al., 2005; Corsetti et al., 2017; Orendorff et al., 2002). Price et al. (2017) compared that ratio between mid-infrared spectra of emissions under different engine conditions for ultra-low sulfur diesel (ULSD) and hydrogenation derived renewable diesel (HDRD) fuels and observed a slightly greater ratio for the ULSD emissions and suggested this was due to the differences in the carbon number distribution of the two fuel emissions. In addition, some other vibrational bands can affect this region through forming overtones and combination bands (Thomas, 2017). Overall, the

absorbance profile in the aliphatic $C-H$ region contains direct and indirect information about carbon number and molecular weight and shows significant variation in laboratory standards and atmospheric samples (Fig. 1) related to their molecular structure. In this work, we adopt a new approach for using mid-infrared spectra to characterize OM. We use variation in aliphatic $C-H$ region to estimate mean carbon number and mean molecular weight of atmospheric samples. In previous studies on the mid-infrared spectrum of atmospheric aerosols, functional group molar abundance in laboratory standards or





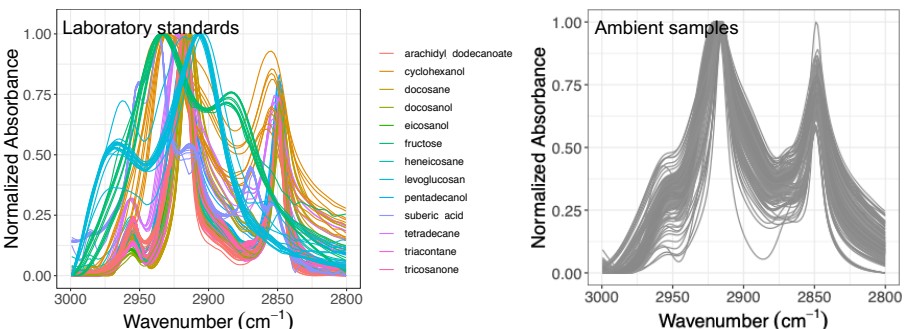

**Figure 1.** Normalized aliphatic C−H spectra of the laboratory standards (left) and several atmospheric samples (right). This figure shows variation in absorbance profile among the standards and atmospheric samples.

total OC from other methods such as TOR were considered as the response variable, while non-normalized absorbances were considered as independent variables (Takahama et al., 2013; Ruthenburg et al., 2014; Reggente et al., 2016). In this manner, linear models resembling the Bougher-Lambert-Beer law were developed. In this study, however, molecular weight and carbon number models were developed using chemical formulas of the laboratory standards (no molar abundance information) and

5 their normalized aliphatic C−H absorbances as independent variables. The current approach extracts detailed information from the mid-infrared spectrum complementary to previous approaches (Fig. 2).

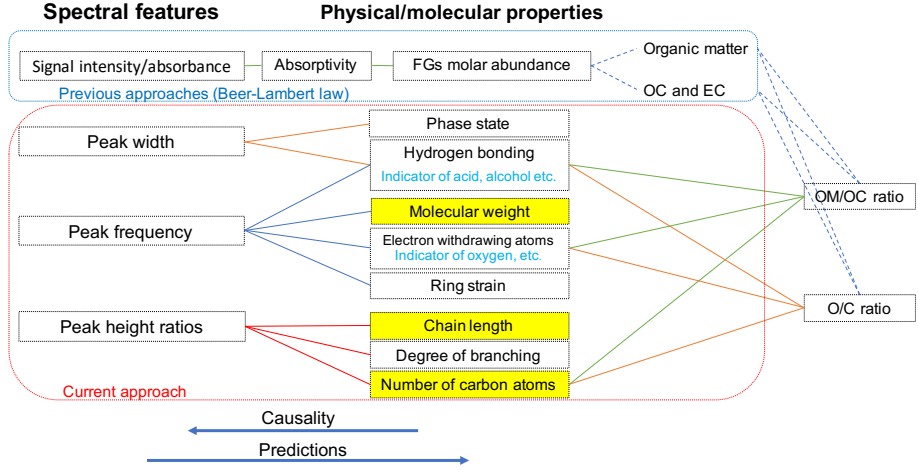

**Figure 2.** Diagram showing the relation between spectral features and molecular/physical properties. The way previous models (for example Ruthenburg et al. (2014); Takahama et al. (2013)) and the current model use spectrum to estimate different parameters is shown in blue and red boxes respectively. Highlighted molecular properties can only be estimated using the current approach.





## 2  Methods

We will describe the atmospheric samples as well as the laboratory standards for the calibration and test set in Sect. 2.1 and 2.2. Thereafter, the methodology for data analysis and interpretation will be discussed in Sect. 2.3, 2.4, and 2.5.

### 2.1  IMPROVE network monitoring sites (sampling and analysis)

Particulate matter with diameter less than 2.5 µm ($PM_{2.5}$) was collected on PTFE filters (25 mm diameter Teflo® membrane, Pall Corporation) every third day at nominal flow rate of 22.8 $L\,min^{-1}$ during 2011 and 2013 at selected sites in the Inter-agency Monitoring of PROtected Visual Environments (IMPROVE) network (http://vista.cira.colostate.edu/improve/). There are, in total, 814 samples collected at 7 sites in the USA in year 2011 and 2161 samples collected at 16 different sites in the USA 2013 (see Fig. 3). 1 out 7 sites in 2011 and 4 out of 16 of sites in 2013 are urban sites and the rest are rural. FT-IR

analysis was performed on the PTFE filters using a Bruker-Tensor 27 FT-IR equipped with a liquid nitrogen-cooled, wideband mercury-cadmium-telluride (MCT) detector, and a resolution of 4 $cm^{-1}$ (data intervals of 1.93 $cm^{-1}$; Nyquist sampling). For samples with low molar abundance of organic compounds, especially aliphatic C−H, baseline correction could not be done properly in the aliphatic C−H region resulting in irregular and negative absorbance profile. These samples were omitted from further analysis and only 798 were analyzed in this work. As can be seen from Fig. 4, data recovery is higher in urban sites than

rural sites due to having a usually more prominent aliphatic C−H peak. Due to this under-sampling, generalizing the results of this work to the whole of rural samples should be done with caution.

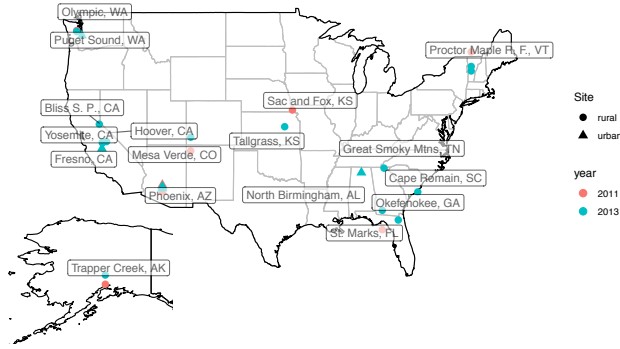

**Figure 3.** The location of IMPROVE sites used for this work (the USA and Alaska); the year at which samples are taken is differentiated by color and the type of the site by point shape.

### 2.2  Laboratory standards (sampling and analysis)

Compounds containing relevant functional groups to atmospheric OM such as aliphatic C−H, alcohol and acid O−H, carbonyl C=O, and with different structures (straight-chain and cyclic) and various chain lengths were used to produce laboratory

standards (Table 1). Five of the compounds used to make laboratory standards were alkanes, just containing aliphatic C−H.

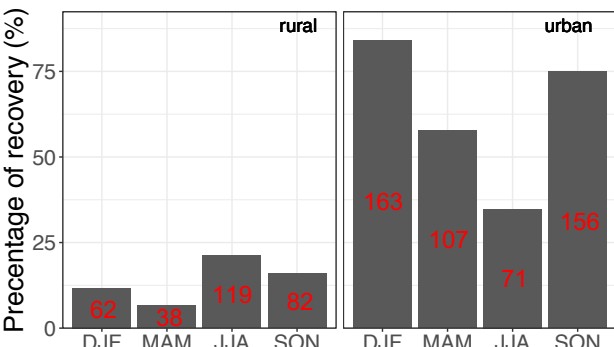

**Figure 4.** Percentage of the samples which were recovered from each category (sample type and season) after baseline correction.

Three were straight-chain alcohols containing alcohol O−H as well. One was cyclic alcohol and one was a cyclic ketone having carbonyl C=O; two were cyclic (not aromatic) sugar derivatives containing several O−H groups. The calibration set also contained an ester, a ketone and one dicaboxcylic acid. All compounds used for creating the standards contained aliphatic C−H which is the main focus of this study. These compounds had comparable absorption coefficients for aliphatic C−H and

the effect of other functional groups and the molecular structure was analyzed indirectly via the change in aliphatic C−H absorbance profile. Some of the laboratory standards and their resulting spectra were taken from Ruthenburg et al. (2014). The rest were created (using a similar protocol) from methanolic solutions with a concentration of $0.1 \, g \, L^{-1}$ and analyzed by FT-IR as follows. Atomized aerosols of the desired compounds were first generated using a TSI Model 3076 Aerosol Generator using the methanolic solutions. Then these particles were conducted by the flow system towards a 47 mm PTFE filter (Pall),

where they were collected. The flow system was composed of a silica gel dryer (for drying the aerosols before collection), a sharp-cut-off 1 μm cyclone and a diluter system (which facilitated the adjustment of aerosol concentration in the line). The pressure drop needed for the flow through the filter was provided by a rotary vacuum pump (Gast 0523-101Q-G588NDX) and the filter flow was controlled by a gas-flow controller (Alicat MCR-100-SLPM-D/5M). The mass on the filters ranged from few micro-grams to tens of micro-grams. After collecting the aerosols on the filters, FT-IR analysis was performed on the PTFE

filters using a Bruker-Vertex 80 FT-IR instrument equipped with a deuterated lanthanum $\alpha$ alanine doped triglycine sulfate (DLaTGS) detector, with the same spectral resolution as the spectra of the ambient samples.

In total, 168 laboratory samples with different composition and molar abundance (absorption amplitude ranging from 0.001 to 2 before normalization) were used from which a subset of 43 samples was kept as a test set and the rest were used as the calibration set. The test set was used solely for the purpose of evaluation of the models developed using the calibration

set. However, the final models, which were applied to ambient samples, were developed using all 168 laboratory standards to increase the model precision.





**Table 1.** Chemicals used in the calibration set to analyze the effect of different physical/chemical properties of organic molecules on aliphatic $C-H$ absorbance profile.

| Compound Name | Formula | Class | Phase State at 25°C | Molecular Weight ($\mathrm{g\,mol^{-1}}$) | OM/OC |
|---|---|---|---|---|---|
| Tetradecane | $C_{14}H_{30}$ | alkane | liquid | 198.4 | 1.18 |
| Hexadecane | $C_{16}H_{34}$ | alkane | liquid | 226.4 | 1.18 |
| Heneicosane | $C_{21}H_{44}$ | alkane | solid | 296.6 | 1.18 |
| Docosane | $C_{22}H_{46}$ | alkane | solid | 310.6 | 1.18 |
| Triacontane | $C_{30}H_{62}$ | alkane | solid | 422.8 | 1.17 |
| 1-Pentadecanol | $C_{15}H_{32}O$ | alkanol | solid | 228.4 | 1.27 |
| 1-Eicosanol | $C_{20}H_{42}O$ | alkanol | solid | 298.6 | 1.24 |
| 1-Docosanol | $C_{22}H_{46}O$ | alkanol | solid | 326.6 | 1.24 |
| Cyclohexanol | $C_6H_{12}O$ | cyclic alcohol | liquid | 100.2 | 1.39 |
| Cyclohexanone | $C_6H_{10}O$ | cyclic ketone | liquid | 98.1 | 1.36 |
| Fructose | $C_6H_{12}O_6$ | Sugars and their derivatives | solid | 180.2 | 2.50 |
| Levoglucosan | $C_6H_{12}O_5$ | Sugars and their derivatives | solid | 162.1 | 2.25 |
| Suberic acid | $C_8H_{14}O_4$ | dicarboxylic acid | solid | 174.2 | 1.81 |
| Arachdyl dodecanoate | $C_{32}H_{64}O_2$ | ester | solid | 480.9 | 1.25 |
| 12-Tricosanone | $C_{23}H_{46}O$ | ketone | solid | 338.7 | 1.23 |

## 2.3 Baseline correction and normalization

The baseline removal is often a useful step in mid-infrared spectroscopy on PTFE filters, like in other methods of spectroscopy. The baseline arises from light scattering by the filter membrane (Mcclenny et al., 1985) and particle as well as electronic transitions of some carbonaceous materials (Russo et al., 2014; Parks et al., 2019). For baseline removal, we used the smoothing spline method on 1500–4000 $\mathrm{cm^{-1}}$ region, where PTFE filter does not absorb, with parameter selection criteria similar to the approach taken by Kuzmiakova et al. (2016). Briefly, a cubic smoothing spline was fitted to the spectrum, and then was subtracted from the raw spectrum to obtain the pure contribution of functional groups at each wavelength. The analyte region (the aliphatic $C-H$ absorption region, 2800–3000 $\mathrm{cm^{-1}}$) was manually excluded from the baseline by setting the weights in this region to zero in the the smoothing spline objective function (refer to Kuzmiakova et al. (2016)). The rest of the spectrum between 1500–4000 $\mathrm{cm^{-1}}$ was included in the baseline by setting the weights one. After baseline correction, the aliphatic $C-H$ absorbances were scaled between zero and one (Fig. 1) for all spectra so that the absorbance profiles were comparable regardless of the absorbance intensity (functional group abundance).





## 2.4 Building the calibration models

We seek the solution of the following linear equation for the calibration models:

$$y = \mathbf{X}b + e, \tag{2}$$

where $\mathbf{X}$ is the normalized spectra matrix, $y$ is vector of response variable (molecular weight or carbon number) and $e$ is a
vector of residuals ($y$ and $\mathbf{X}$ are assumed to be centered). In spectroscopic applications, due to indeterminacy (more independent variables than the number of samples) and collinearity (inter-correlation between independent variable) the ordinary least squares (OLS) method is not applicable or is not robust unless regularized. Among the common methods developed for treating such a data structure, we chose univariate ($y$ is a vector, i.e. has one variable) partial least squares regression (PLSR) for this work (Wold et al., 1983). Univariate PLSR projects $\mathbf{X}$ onto $\mathbf{P}$ basis with orthogonal scores $\mathbf{T}$ and residual matrix $\mathbf{E}$ such
that, the covariance between each score column and $y$ is maximized (in each step of deflation). In Eq. (4), $c$ is the regression coefficient of $y$ as a function of scores ($\mathbf{T}$) and $f$ is the vector of residuals.

$$\mathbf{X} = \mathbf{T}\mathbf{P}^{\top} + \mathbf{E}, \tag{3}$$

$$y = \mathbf{T}c + f. \tag{4}$$

After solving the PLS problem for candidate models with different number of latent variable (LVs), we ran a repeated 10-fold
cross validation on candidate models to indicate the number of latent variables giving the minimum RMSE for the calibration set. Thereafter, a simpler model (with fewer LVs) whose RMSE was no more than one standard error above the model with minimum RMSE was chosen (Hastie et al., 2009). The optimal number of LVs for molecular weight and carbon number models was 19 and 20 respectively.

## 2.5 Interpreting the models using the basic features

Although the models have considerably fewer LVs (approximately 20) than the original wavenumbers (105), the lack of physical interpretability and remaining number of LVs still hinders physical interpretation of the models. Therefore, we first analyze the basic (physically interpretable) features of the mid-infrared spectrum -peak frequencies, widths and ratios in aliphatic $\mathrm{C-H}$ region- for the calibration set and their relation with carbon number and molecular weight (Sect. 3.1). Spatial and temporal variation of these patterns in atmospheric samples are also analyzed and related to similar patterns in laboratory standards.

25       The four basic features of the ambient sample spectra were used to build a classification and regression trees (CART) (Breiman et al., 1983) to approximate the PLS predictions of mean molecular weight and carbon number and better understand their connection with the underlying spectral absorption characteristics. In this approach, binary decision trees are generated to the classify the PLS estimates based on partitioned domains of their basic spectral features. The CART algorithm expands the trees in the order of decreasing explanatory power until certain stopping conditions (minimum number of observations in
terminal nodes or minimum improvement of explanatory power at each step of splitting) are satisfied.



# 3   Results and discussions

First, the basic features of aliphatic $C-H$ profile are discussed in atmospheric and laboratory samples followed by a similarity check between the two (Sect. 3.1). Then, development of quantitative models for predicting molecular weight and carbon is described, followed by investigation of their performance in the calibration and test (Sect. 3.2). Thereafter, estimates of the

models are discussed for atmospheric samples and compared with the results reported in literature (Sect. 3.3). Finally, the basic features introduced earlier are used to classify the results of the sophisticated models in order to obtain a better understanding of the way the models function (Sect. 3.4).

## 3.1   Basic features

Basic features of the spectrum in aliphatic $C-H$ region were calculated for atmospheric samples and laboratory standards

to study their temporal and spatial variation and their relation with molecular properties such as molecular weight, carbon number, and the OM/OC ratio. These variables although few, can give a good estimate of the absorbance profile and make it more interpretable.

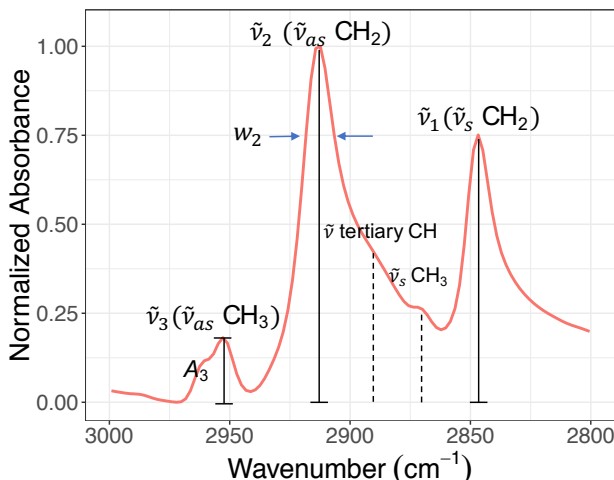

**Figure 5.** A sample $C-H$ spectrum showing the convention of peak parameters used in this study. The symmetric $CH_2$ ($\tilde{\nu}_s$ $CH_2$) wavenumber is denoted by $\tilde{\nu}_1$. The asymmetric $CH_2$ ($\tilde{\nu}_{as}$ $CH_2$) wavenumber is denoted by $\tilde{\nu}_2$ and the asymmetric $CH_3$ ($\tilde{\nu}_{as}$ $CH_3$) wavenumber by $\tilde{\nu}_3$. Absorbance and width of the $ith$ peak are also denoted by $A_i$ and $w_i$, respectively.

Figure 5 shows the convention of spectral features in the aliphatic $C-H$ (2800–3000 $\mathrm{cm}^{-1}$) region used in this study. Apart from methine group (tertiary $C-H$), which has a very weak absorption (Pavia et al., 2008), there are two doublets in this

region corresponding to $CH_2$ and $CH_3$ symmetric and asymmetric stretching vibrations. The $CH_3$ symmetric peak is typically suppressed by the surrounding peaks and is not completely distinguishable. Among the remaining peaks, the symmetric $CH_2$ ($\tilde{\nu}_s$ $CH_2$) wavenumber is denoted by $\tilde{\nu}_1$. Likewise, the asymmetric $CH_2$ ($\tilde{\nu}_{as}$ $CH_2$) wavenumber is denoted by $\tilde{\nu}_2$ and the



asymmetric $CH_3$ ($\tilde{\nu}_{as}$ $CH_3$) wavenumber by $\tilde{\nu}_3$. Absorbance and peak width of the $ith$ peak are also denoted by $A_i$ and $w_i$, respectively.

In the next chapters, the variation of the mentioned spectral features are studied in laboratory standards and atmospheric samples. For this purpose, atmospheric samples are separated in to urban, rural and burning samples. The burning category

constitutes 95 samples of urban or rural sites and is taken from clusters 9a, 9b and 10 of Bürki et al. (2019) based on their spectral similarity. These samples are believed to be influenced by residential wood burning or wildfires since they were usually taken during a known fire period (Rimfire in California in 2013) or during winter months when residential wood burning typically occurs.

### 3.1.1 Asymmetric $CH_2$ peak wavenumber ($\tilde{\nu}_2$)

We calculated the second peak wavenumber ($\tilde{\nu}_2$) for laboratory standards and atmospheric samples using a simple peak finding algorithm based on the first and second numerical derivatives of the spectrum. Generally, for laboratory standards the frequency decreases with increasing molecular weight until it reaches an asymptotic state after $200 \, \mathrm{g \, mol^{-1}}$ (Fig. 6). The curve in Fig. 6 shows the theoretical peak frequency of the aliphatic $C-H$ when the bond spring constant is assumed to be $10^3 \, \mathrm{N \, m^{-1}}$ (Pavia et al., 2008), and the reduced mass is calculated based on a ball-and-string assumption composed of the hydrogen atom

(first "ball") and the rest of molecule (second "ball"). The only effect considered in this model is the variation of the reduced mass of the oscillator. The fact that the less-oxygenated laboratory samples follow the theoretical line closely implies that the value of the spring constant considered here is, on average, a good approximation. However, especially for highly oxygenated (high OM/OC ratio) molecules and those with in liquid phase (which have lower molecular weight), the absorption frequency deviates from the theoretical line (higher frequency) due to higher levels of inter-molecular interaction.

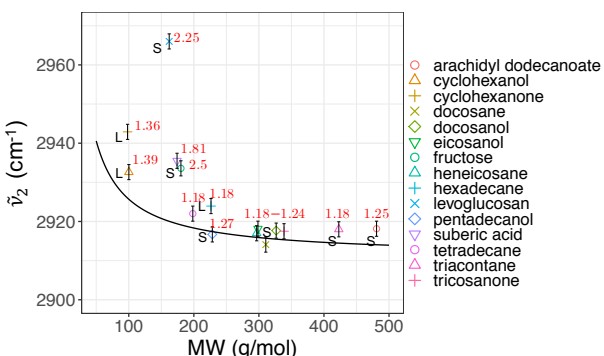

**Figure 6.** Scatter plot showing the variation of the second peak wavenumber ($\tilde{\nu}_2$) with molecular weight (MW) in the calibration set, affected by the OM/OC ratio and phase state. The black line shows the theoretical frequency with a spring constant equal to $10^3 \, \mathrm{N \, m^{-1}}$ for all $C-H$ bonds. The OM/OC ratio and phase state are shown for the samples. The error bars show uncertainty in calculated peak frequency due to FT-IR scan resolution.

Regarding the atmospheric samples, most of categories have a peak density in 2915–2925 $cm^{-1}$, close to that of straight-chain molecules of the laboratory standards (Fig. 7, first row). Urban samples have a wider shoulder on the right side (around 2925 $cm^{-1}$) in summer when the samples are expected to be more aged. Other variations are believed to be insignificant considering the scan resolution of the FT-IR instrument.

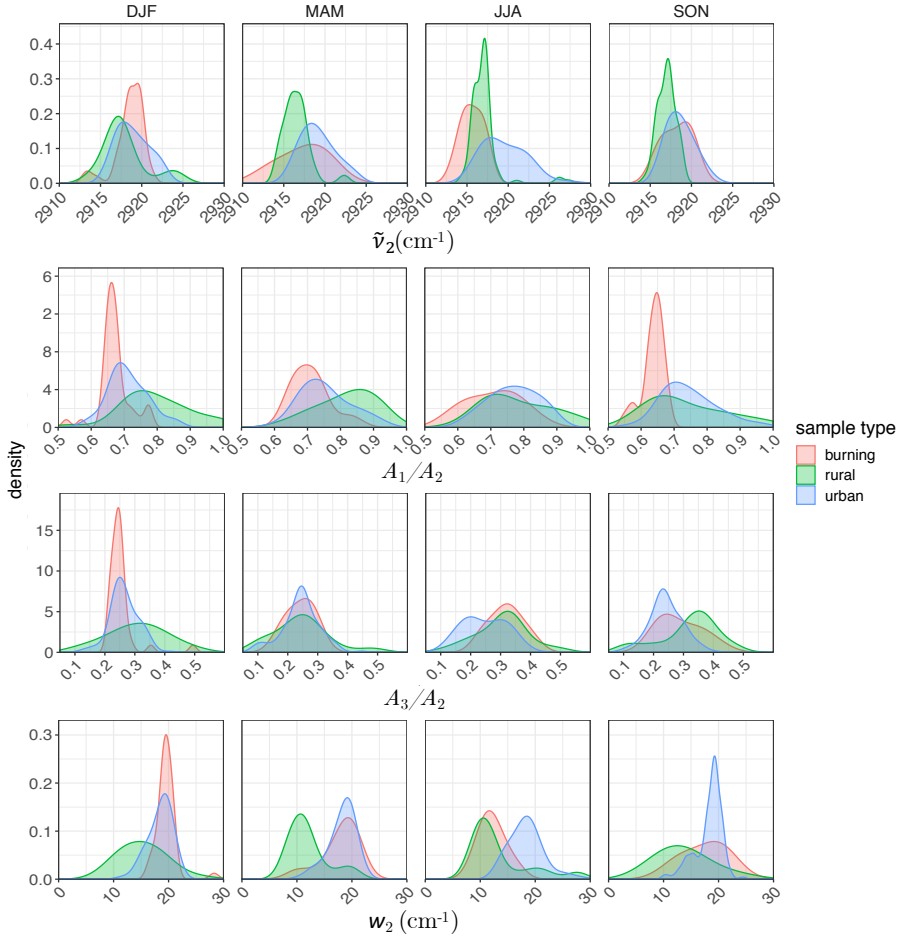

**Figure 7.** Kernel density estimate of second peak wavenumber ($\tilde{\nu}_2$), the ratio of peak heights of symmetric $CH_2$ to asymmetric $CH_2$ stretching ($A_1/A_2$), the ratio of peak heights of asymmetric $CH_3$ to asymmetric $CH_2$ stretching ($A_3/A_2$), and second peak width ($w_2$) of aliphatic $C-H$ band in mid-infrared spectra of atmospheric samples segregated based on sample type and season.

5   **3.1.2   Peak height ratios ($A_i/A_2$)**

Analyzing the laboratory standards shows that a relatively linear but scattered relation exists between carbon number and the $A_1/A_2$ ratio in the calibration set (Fig. 8, upper panel). Suberic acid that is the only dicarboxylic acid in the laboratory





standards that does not follow the general trend, probably due to strong dimerization. As mentioned in Sect. 1.2, the $A_1/A_2$ ratio compares symmetric and asymmetric absorbance of methylene functional group and its connection with carbon number has already been highlighted in FT-IR analysis of some types of diesel fuels (Price et al., 2017). Increase in $A_1/A_2$ is also observed between solid and liquids, consistent with the work of Corsetti et al. (2017). We also observe a nonlinear relation

between the $A_3/A_2$ ratio and carbon number with different levels based on branching and terminal functionalization (Fig. 8, lower panel). This ratio is equal to zero for molecules lacking methyl group such as simple cyclic molecules while increasing as the number of branches containing terminal methyl increases.

Results show a clear separation in atmospheric samples regarding the sample type and season for both $A_1/A_2$ and $A_3/A_2$ ratios (Fig. 7, second and third row). The samples influenced by burning usually have the lowest $A_1/A_2$ ratio (Fig. 7, second

row). This observation is consistent with the presence of molecules with longer chains, as observed for laboratory samples. Bürki et al. (2019) showed that urban samples (in the same dataset) have their highest average OM/OC ratio in summer which is concurrent with the their highest $A_1/A_2$ ratio which suggests shorter chain length. The highest $A_1/A_2$ ratio for rural samples is observed in spring when the aerosols are highly oxidized (Bürki et al., 2019). This suggests that aged aerosols have lower carbon number probably due to the fragmentation process. The measured $A_1/A_2$ ratio for majority of the atmospheric samples

ranges between 0.6 to 0.8, which is consistent with the value for laboratory standards. Results also show that the $A_3/A_2$ ratio is higher in rural samples compared to urban samples (with the exception of spring) suggesting a higher $CH_3$ to $CH_2$ abundance in those samples. This observation can be due to lower carbon number or higher number branches containing $CH_3$. Like the $A_1/A_2$ ratio, we observe fewer samples with low $A_3/A_2$ ratios in urban sites in summertime. The $A_3/A_2$ ratio falls between 0.1–0.4 for majority of atmospheric samples, which is consistent with the value for laboratory standards. It is worth noting that

peaks in atmospheric samples are more overlapped than laboratory standards, which makes calculation of peak ratios based on extrema of the original spectra imprecise. In order to obtain peak ratios precisely, a peak fitting method based on Gaussian peaks was applied to atmospheric samples.

### 3.1.3   Peak width ($w_i$)

We observe a clear correlation between $w_2$ and the OM/OC ratio in the calibration set when solid and liquid phases are

considered separately. As mentioned in Sect. 1.2, hydrogen bonding increases the peak width, and the extent of hydrogen bonding is usually a good indicator of the OM/OC ratio. This is because hydroxyl, hydroperoxyl, and carboxyl groups that form hydrogen bonds are among the most effective functional groups in SOA formation due to the significant vapor pressure reduction they cause (Seinfeld and Pandis, 2016). In this study, $w_2$ is defined as the peak width at 75 % of the maximum amplitude. This position is chosen for robustness of the measurement algorithm (to avoid interference with other peaks);

however, it can be converted to full width at half maximum (FWHM) assuming the proper peak profile ($w_2$ is 65 % of FWHM for a Gaussian peak). In addition to hydrogen bonding and phase state, superposition of a multitude of peaks with slightly different profiles can also have a statistical positive or negative effect on the peak width in mixtures (see Supplement Sect. S1). The observed peak width in atmospheric sample spectra is the result of all above-mentioned factors. However, since all laboratory standards are produced with pure compounds, the significance of mixture effect cannot be evaluated.





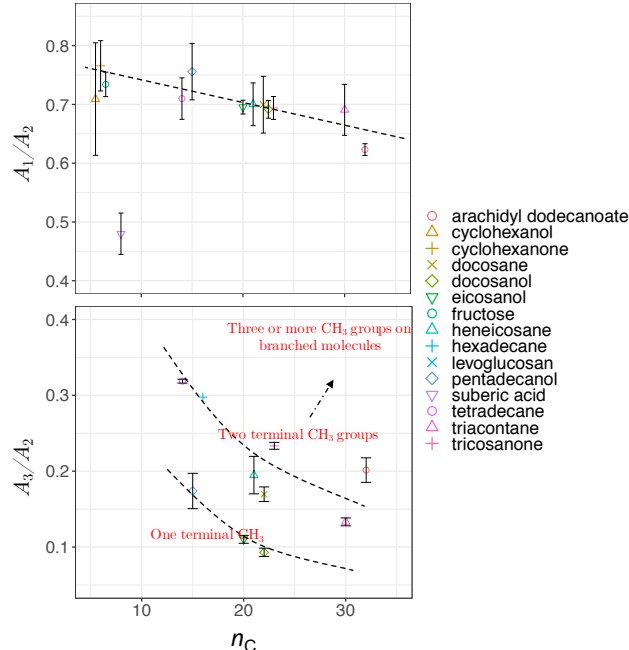

**Figure 8.** Scatter plots showing the relation between carbon number ($n_C$) and the ratio of peak heights of symmetric $CH_2$ to asymmetric $CH_2$ stretching ($A_1/A_2$, upper panel), and the ratio of peak heights of asymmetric $CH_3$ stretching to asymmetric $CH_2$ stretching ($A_3/A_2$, lower panel), averaged for each substance in laboratory standards. Error bars show $\pm$ one standards error from the average and dashed lines are visual guides for the trends and levels.

Figure 7 (fourth row) shows a distinct distribution of $w_2$ considering spatial and temporal variations as well as sample type. Rural samples have a smaller value of $w_2$ compared to urban and burning samples, although the former are usually more oxidized (have higher OM/OC ratio). This observation suggests that other factors such as phase state and statistical effects likely outweigh the oxygenation effect on absorption peak width.

### 3.1.4 Spectral similarity (dimension reduction)

In previous sections, the basic features of spectra in the aliphatic $C-H$ region were presented and discussed for atmospheric samples and laboratory standards. Here, we check the spectral similarity between atmospheric complex mixtures and laboratory pure standards by means of principal component analysis (PCA), before developing quantitative models.

The spectral data of laboratory standards are highly collinear as can be seen from their correlation matrix heat map (Fig. A1). In this case, PCA is efficient for reducing the data dimension so that only the first 6 principal components (PCs) explain around 99 % of variance in the spectra (Table 2). For the sake of comparison, we have projected the spectra of atmospheric samples onto the 6 PCs. The results show that their scores, when projected onto laboratory PCs, are surrounded by laboratory standards. Many spectra, particularly urban ones, are clustered close to tetradecane for the first 4 PCs (Fig. 10); greater differentiation is





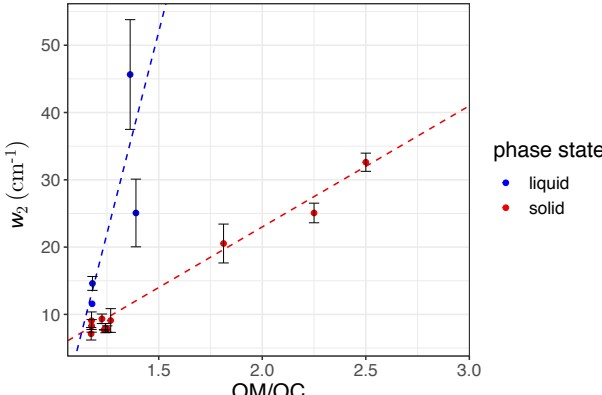

**Figure 9.** The average value of second peak width ($w_2$) measured for each compound in the calibration set versus the OM/OC ratio, colored based on compound phase state at laboratory condition (25 °C). Error bars show $\pm$ one standards error from the average and dashed lines are visual guides.

found among the higher PCs. This observation suggests that laboratory standards are able to capture the main variations in the spectra of atmospheric samples, which have a more regular shape close to that of straight-chain alkanes.

**Table 2.** Importance of the first 6 principal components in the laboratory standards.

|  | PC1 | PC2 | PC3 | PC4 | PC5 | PC6 |
|---|---|---|---|---|---|---|
| **Standard Deviation** | 1.414 | 0.668 | 0.647 | 0.332 | 0.203 | 0.133 |
| **Proportion of Variance** | 0.651 | 0.145 | 0.136 | 0.036 | 0.014 | 0.006 |
| **Cumulative Proportion** | 0.651 | 0.796 | 0.932 | 0.968 | 0.982 | 0.988 |

## 3.2 Developing and evaluating the models

PLS with cross validation was used to develop quantitative models for molecular weight (MW) and carbon number ($n_C$) with
5 the calibration set composed of 143 samples including all compounds over the available mass range. The OM/OC ratio was then calculated from those two parameters ($OM/OC = \frac{MW}{12.01 n_C}$). The developed models gave reasonably good fit results ($r^2$ ranging from 0.94 to 0.99) for molecular weight, carbon number, and indirect OM/OC ratio in the calibration set (Figure 11).

The prediction ability of the PLS models was then evaluated using a test set composed of 43 samples which were not used for developing the models. Models also performed reasonably well in predicting molecular weight, carbon number and OM/OC
10 ratio in the test set with $r^2$ ranging from 0.92 to 0.98 (Fig. 11). The predictions with high relative error were attributed to laboratory samples with low molar abundance (low signal-to-noise ratio), for which the baseline correction had the highest uncertainty. The is not a concern when applying the models to atmospheric samples since the atmospheric samples with low signal-to-noise ratio were omitted in the first step (Sect. 2.1).

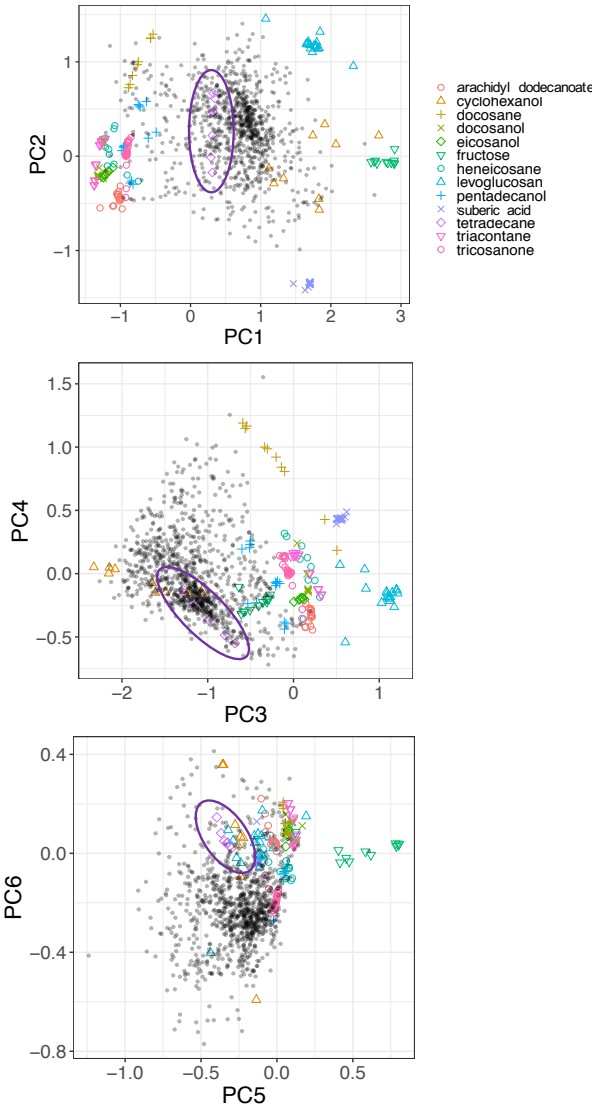

**Figure 10.** Bi-plots showing the scores of normalized spectra of laboratory standards (color) and normalized spectra of atmospheric samples (black) projected onto the first 6 principal components (calculated for laboratory standards). Purple ellipses indicate the location of tetradecane standards.

## 3.3 Applying the models to atmospheric samples

After checking the performance of the models on the calibration and test set, we used all laboratory standards to produce PLS models to applying to ambient samples. In the following sections, the estimates of OM/OC, mean molecular weight, and mean carbon number for ambient samples are shown in different categories based on season and sample type (rural, urban

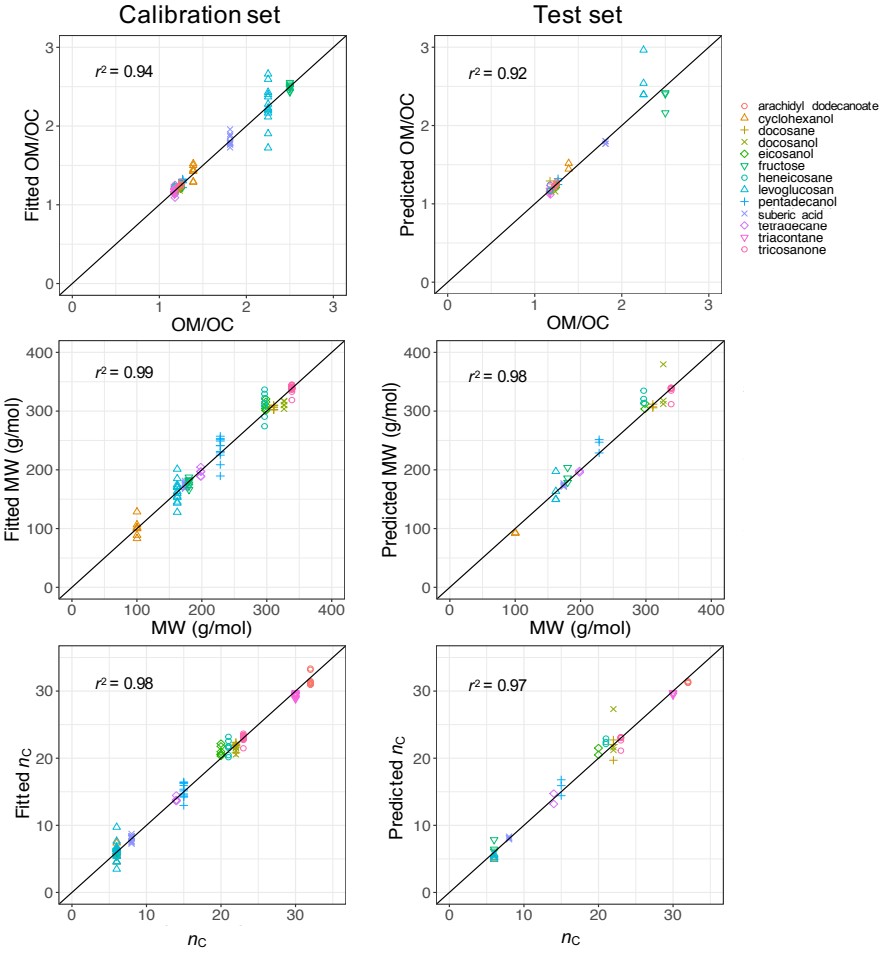

**Figure 11.** Scatter plot of fitted (predicted) indirect OM/OC ratio, molecular weight (MW), and carbon number ($n_C$) against the values from chemical formula of the calibration set (test set). The diagonal black lines indicate perfect fit (1:1).

and burning) after omitting the physically unreasonable values. Thereafter, the trends and absolute values are compared with previous studies (when available) and our expectations based on aging process and aerosol emission sources.

In this work, we have assumed that we can obtain mean mixture (atmospheric samples) properties from the normalized spectrum of of the mixture using the models developed for pure compounds (laboratory standards). This assumption relies on the linearity of the property estimation models (which is consistent with our calibrations, Eq. (4)), and equality of the absorption coefficients of the compounds existing in the mixture (see Appendix B for more information). Thus, the absorption coefficient of aliphatic C–H has been assumed to be relatively similar between the compounds existing in atmospheric samples. Although the aliphatic C–H absorption coefficients of the laboratory standards were similar in this study, the variability of this absorption coefficient is relatively less-studied for compounds existing in atmospheric OM and needs to be addressed in the future. This



assumption is a potential source of error that may change the accuracy of the results, but the estimates for atmospheric samples shown in the following sections suggest that this assumption does not overwhelm the findings.

### 3.3.1 OM/OC ratio

The OM/OC ratio is the first parameter that we investigate here since it has been studied extensively in atmospheric aerosols
(Bürki et al., 2019; Hand et al., 2019; Ruthenburg et al., 2014; Takahama et al., 2011; Simon et al., 2011; Aiken et al., 2008). Moreover, it can be used as an indirect evaluation for mean molecular weight and mean carbon number estimates as the indirect OM/OC ratio is calculated from those two. An indirect OM/OC estimate that is consistent with previous studies implies that estimates of molecular weight to carbon number are also likely to be reasonable.

The OM/OC ratio is estimated to be generally lower for urban samples ($\approx 1.5$) than rural samples ($\approx 1.8$; Fig. 14, first
row). The lower OM/OC ratio in urban sites is thought to be related to emission sources that are generally hydrocarbon, with low OM/OC ratio emitted from gasoline and diesel vehicles (fuel combustion and unburned motor oil) as a major part of anthropogenic SOA precursors (Gentner et al., 2012)) as well as cooking. These organic molecules do not undergo significant oxidation and aging as the monitoring sites are generally close to the emission sources. In contrast, organic aerosols usually undergo several steps of oxidation and receive substantial condensation of oxidized vapors, which results in higher OM/OC
ratio at rural and remote sites. Previous studies using several different methods (including FT-IR and AMS) show the same trend in urban and rural sites (Ruthenburg et al., 2014; Zhang et al., 2007; Simon et al., 2011; Bürki et al., 2019). In addition, the majority of the samples are in the range that is usually considered for OM/OC ratio, i.e.,1.4–1.7 (Russell, 2003). We also observe that samples influenced by burning, especially residential wood burning, have lower OM/OC ratio ($\approx 1.4$) than those associated with more oxidized aerosol such as rural site, which has also been estimated by Bürki et al. (2019).

The OM/OC ratio in urban sites is estimated to be higher in summer compared to other seasons, especially winter (Fig. 14, first row) which is believed to be caused by more intense photochemical aging in summertime (Kroll and Seinfeld, 2008). In rural sites, the trend becomes more complicated as vegetation, as major biogenic SOA emission sources, is more active in summer time (Yuan et al., 2018; Seinfeld and Pandis, 2016). Samples influenced by burning are also estimated to have higher OM/OC in summer when samples are affected by wildfires compared to winter when samples are mostly affected by residential
wood burning. However, the contribution of photooxidation relative to emission sources is not clear in this case as they are coupled in these observations (Bürki et al., 2019).

In order to have a direct comparison with other methods, we chose the Phoenix, AZ, monitoring site, for which recovery percentage of the baseline correction method is close to 100 %, and compared our indirect OM/OC ratio estimates to the corresponding ones, calculated by Bürki et al. (2019). The latter method uses molar abundance information of functional groups
in laboratory standards in addition to a much wider region of non-normalized mid-infrared spectrum (1500–4000 $cm^{-1}$). The median seasonal OM/OC ratio of this study underpredict that of Bürki et al. (2019) by 0.12 on average, while reproducing the same temporal trends. Some of the discrepancies may be due to insensitivity of spectral features to molecular characteristics in certain domains — for instance, the variation of peak frequency $\tilde{\nu}_2$ diminishes with increasing molecular weight (Sect. 3.1.1).





However, the overall agreement between the two methods is reasonable considering the indirect nature of estimates in our work (Fig. 12).

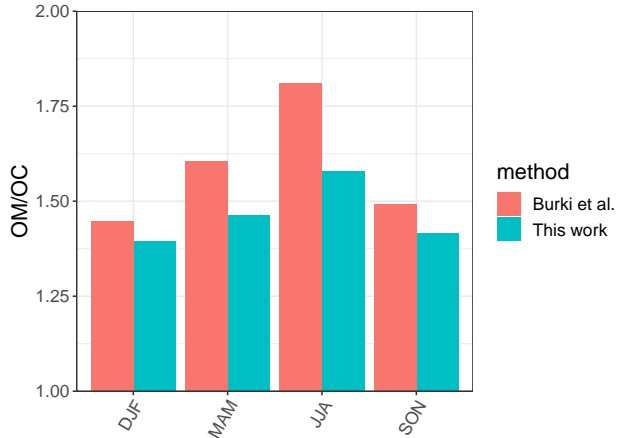

**Figure 12.** Bar chart showing median OM/OC ratio calculated for each season based on samples collected in the Phoenix, AZ, monitoring site using our method and the one used by Bürki et al. (2019).

### 3.3.2 Molecular weight (MW)

The PLS model estimates the mean molecular weight to range between 100–350 $g\,mol^{-1}$ for majority of the samples (Fig. 14, second row). To best of authors' knowledge no extensive study has been performed on mean molecular weight of ambient organic aerosol constituents. Nevertheless, the estimated range is reasonably close to that of the studies that have been done. Those studies measured molecular weights up to 200 $g\,mol^{-1}$ for SOA constituents using GC/MS and ion chromatography (Cocker III et al., 2001; Jang and Kamens, 2001b; Kalberer, 2004), an average molecular weight between 200–300 $g\,mol^{-1}$ for atmospheric HUmic-LIke Substances (HULIS) using electro-spray ionization (ESI) (Graber and Rudich, 2006), and an average molecular weight between 300–450 $g\,mol^{-1}$ for oligomers formed in a smog chamber, measured using laser desorption/ionization mass spectrometry (LDI-MS) (Kalberer et al., 2006). Although particle-phase oligomerization processes result in high-MW compounds (Jang and Kamens, 2001a; Tolocka et al., 2004; Shiraiwa et al., 2014), the abundance of these compounds is usually debated since the available experimental results regarding the reversibility of accretion reactions are contradictory (Kroll and Seinfeld, 2008). Moreover, oligomer formation may be overestimated in laboratory conditions compared to atmospheric particles (Kroll and Seinfeld, 2008; Kalberer, 2004; Trump and Donahue, 2014).

Our model estimates lower mean molecular weight for rural samples ($\approx 200$ $g\,mol^{-1}$) compared to urban ones ($\approx 240$ $g\,mol^{-1}$), while burning samples are estimated to constitute the heaviest molecules ($\approx 290$ $g\,mol^{-1}$). This observation is consistent with our knowledge of emission sources. Emissions in urban areas are influenced by long-chain hydrocarbons from combustion products and motor oil (Gentner et al., 2012), while biomass burning is accepted to be the primary source of





high-MW HULIS (Li et al., 2019). We also observe a decrease in mean molecular weight peak density in urban samples from winter to summer that is believed to be attributed to fragmentation during more intense photooxidation in summer Hand et al. (2019); Jimenez et al. (2009), for emission sources that do not change drastically between the two seasons. The same phenomenon is observed in LDI mass-spectra of some urban samples in summer and winter reported by Kalberer et al. (2006).

Although the reduction in mean molecular weight due to fragmentation can be compensated for by addition of heavy atoms to the molecule during oxidation, our results suggest that the overall direction of photooxidation in urban sites is reduction of the mean molecular weight.

### 3.3.3   Carbon number ($n_\mathrm{C}$)

The PLS model estimates that the recovered rural samples usually have lower mean carbon number compared to urban samples

and the samples influenced by burning (Figure 14, third row). Higher mean carbon number estimates in urban sites (highest probability density around 16), which are coincident with high elemental carbon (EC) values from TOR measurements (Fig. C1), can be attributed to major EC sources such as combustion of fossil fuel and biomass. This is also consistent with high SOA formation potential of molecules with 15–25 carbon in diesel fuel shown by Gentner et al. (2012). Samples affected by burning are estimated to have the highest mean carbon number among all samples. This observation is consistent with the emissions of

plant cuticle waxes, mainly composed of straight-chain hydrocarbons, observed during biomass burning (Hawkins and Russell, 2010) as well as HULIS (Graber and Rudich, 2006). We also observe a decrease in estimated mean carbon number of urban samples from winter to summer suggesting fragmentation during aging and photooxidation processes.

The carbon-oxygen estimates of the PLS models are consistent with the existing numerical simulation. We compared our estimates with the numerical simulations by Jathar et al. (2015). Multi-generational oxidation model used by Jathar et al.

(2015) (Statistical Oxidation Model, SOM) in a 3-D air quality model for simulating SOA in Los Angeles and Atlanta (two urban locations) shows that carbon number in SOA ranges from 3 to 15 with the concentration peaks around 7, 10 and 15 (Fig. 13). For this comparison, we calculated the carbon-oxygen grid from our molecular weight and carbon number estimates, assuming the organic molecules have a chemical formula of $\mathrm{C}_{N_c}\mathrm{H}_{2N_c+2-N_o}\mathrm{O}_{N_o}$ (a common assumption and one used by Jathar et al.). Our models for IMPROVE network estimate mean carbon number peaks (number density) for rural, urban, and

burning samples to be around 8, 16 and 18 respectively, while the total range is limited to 3–19 (Fig. 13). We also estimate the oxygen number to range from 2 to 6 for the majority of the samples. It should be noted that this is as an order of magnitude comparison since the time frame and the location of the two studies are different and the numerical simulation by Jathar et al. (2015) only considers SOA.

### 3.4   Model interpretation

Reducing the spectrum to four basic features introduced in Sect. 3.1 ($\tilde{\nu}_2$, $A_3/A_2$, $A_1/A_2$, $w_2$) is a manual data compression onto a basis set of interpretable variables. Though information loss is inevitable, it was shown in Sect. 3.1 that these basic features are still sufficient for qualitative explanation of spectral variations associated with different emission source and aerosol aging





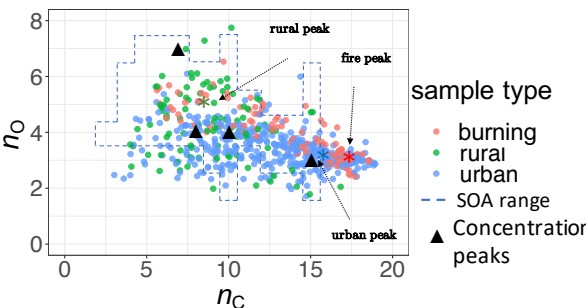

**Figure 13.** Comparison between carbon-oxygen grid simulated by Jathar et al. (2015) for Atlanta and Los Angeles with sample points estimated for IMPROVE network (2011 and 2013) from the molecular weight and carbon number estimates of this study. The dashed lines show the range of simulated carbon and oxygen and the triangles indicate the location of the highest SOA concentrations for simulation of Jathar et al. (2015).

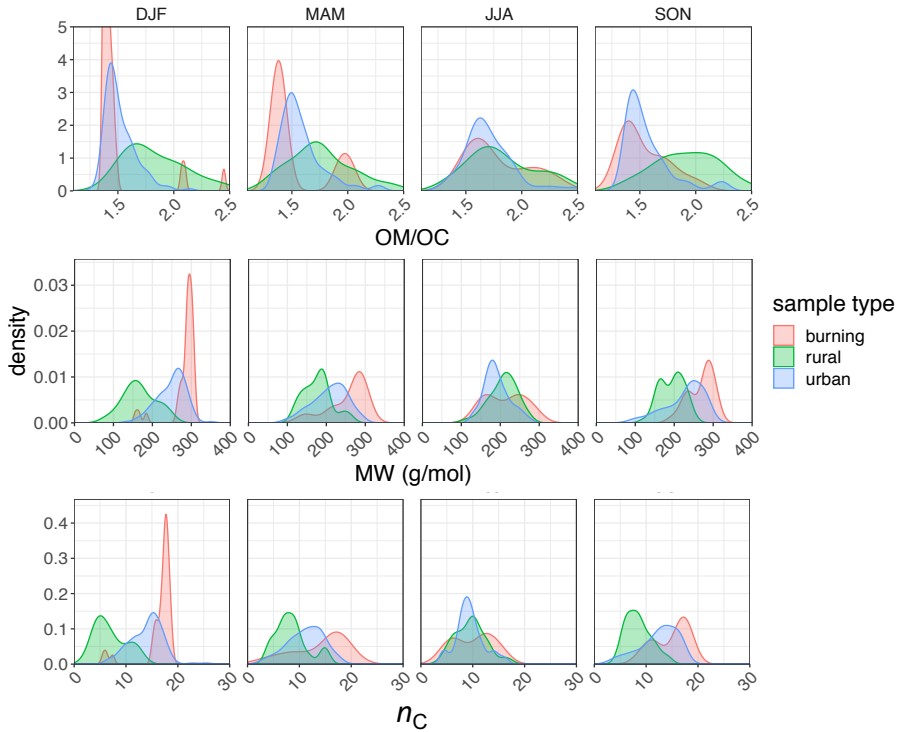

**Figure 14.** Kernel density estimates of indirect OM/OC ratio, molecular weight (MW) and carbon number ($n_C$) estimated from normalized aliphatic C–H mid-infrared absorbances by PLS models (segregated by sample type and season).

process. In this section, predictions made by the PLS models on ambient samples are grouped based on the four basic features using CART (Fig. 15) in order to form a better understanding of how the sophisticated PLS models function.





The regression trees show that the peak ratios are observed to be the main grouping parameter for both carbon number and molecular weight (Fig. 15). The inverse relation of peak ratios with carbon number appears in most of the splitting nodes of carbon number and molecular weight regression trees (Fig. 15). This is consistent with the observed relation between carbon number and peak ratios in the calibration set (Fig. 8). Assuming that molecular weight is highly correlated with carbon number,

the classification of molecular weight based on peak ratios is also expected. The peak frequency ($\tilde{\nu}_2$) appears once as a node in molecular weight tree and classifies the estimates based on the same trend that was observed in the calibration set (Fig. ??). The second peak width ($w_2$) also appears few times in the nodes probably adding information about the OM/OC ratio and phase state. The two trees shown explain only around 50 % of the variation of estimates made by the PLS models. The explained variation can be increased to an arbitrarily high number through the use of more branches in the fitting data set, but

the predictive capability of regression trees for new samples depends highly on their similarity to the training set.

In summary, regression trees show that the predictions of the models are generally consistent with the observed trends of the basic features in the calibration set (Supplement Sect. S2 supports this conclusion for individual spectra for which the PLS models estimate quite different parameters). This observation implies that the PLS predictions of carbon number and molecular weight are not independent of these basic features. However, the sophisticated PLS models use other fine features in addition

to mentioned basic features to extract more detailed information and to reduce variabilities stemming from different sources such as baseline correction.

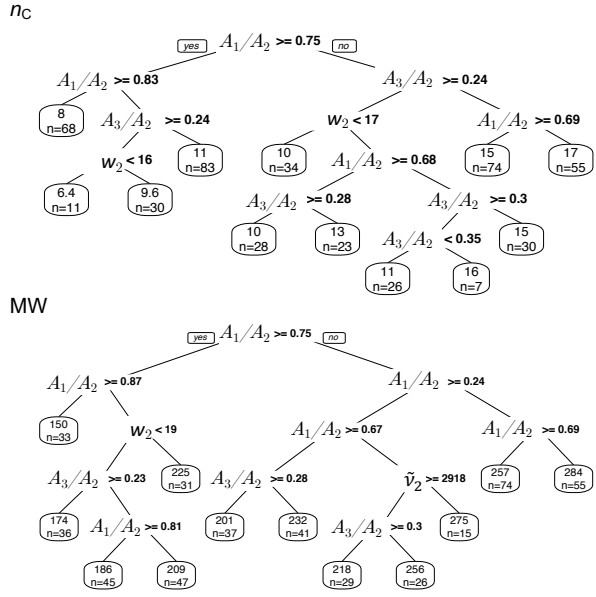

**Figure 15.** Regression tree of molecular weight (MW) and carbon number ($n_C$) estimates in atmospheric samples based on basic spectral features: second peak frequency ($\tilde{\nu}_2$), the ratio of peak heights of symmetric $CH_2$ stretching to asymmetric $CH_2$ stretching ($A_1/A_2$), the ratio of peak heights of asymmetric $CH_3$ to asymmetric $CH_2$ stretching ($A_3/A_2$) and second peak width ($w_2$) of aliphatic $C-H$ band.



## 4 Concluding remarks

Normalized aliphatic C−H profiles in mid-infrared spectrum were used in this study to estimate carbon number and molecular weight of atmospheric OM. First, it was shown that the spectral features in this region such as peak frequencies and ratios, are correlated with carbon number, molecular weight, and the OM/OC ratio for laboratory standards. We also observed a

meaningful temporal and spatial variation of those features in atmospheric aerosol samples. Thereafter, PLS models were developed on laboratory standards to estimate the mentioned parameters in atmospheric aerosol samples. Estimated molecular weight and carbon number reconstruct the OM/OC values in ambient aerosol that are consistent with previous studies with a reasonable difference (an average undeprediction of 0.12). These new models estimate lower mean carbon number and mean molecular weight in more aged aerosols of the same source highlighting the fragmentation role in aging process (Murphy et al.,

2012). Moreover, they estimate relatively less oxidized, heavier molecules with higher carbon number for samples influenced by burning. The findings show that the new technique can help us better understand characteristics of OM due to source emissions and atmospheric processes. In addition, as carbon number and molecular weight are important characteristics used by recent models (Shiraiwa et al., 2017a; Li et al., 2016; Pankow and Barsanti, 2009; Kroll et al., 2011; Donahue et al., 2011) to describe evolution in OM composition, this technique can provide semi-quantitative, observational constraints on these

variations at the scale of the network as well as for laboratory experiments.

Only around 27 % of the existing samples could be analyzed with the developed models due baseline correction limitations posed by low OM mass on the filters. Under-sampling is more severe in rural sites although expected trends (such as higher OM/OC ratio) are observed even in the current subset. As a result, one should be cautious when extending the results of this study to draw general trends. Finally, although some inaccuracy in the results is likely due to extrapolating from laboratory

standards and the indirect nature of the models (for which more research is needed), estimates of molecular weight, carbon number, and the OM/OC ratio were shown to be reasonable. Further evaluation with different molecules and molecular mixtures can better constrain these estimates.





## Appendix A: Correlation matrix heat map

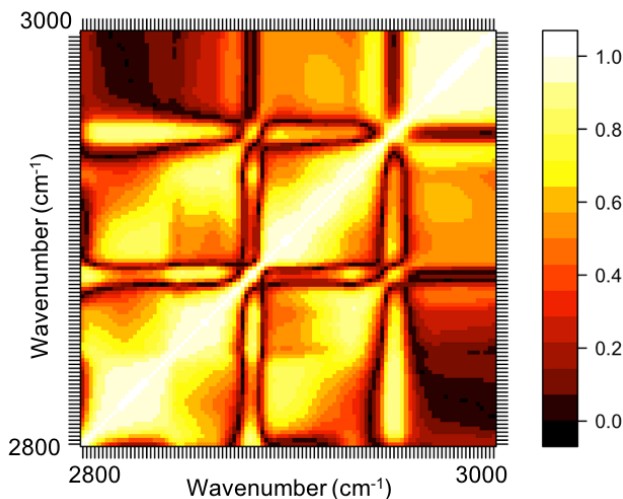

**Figure A1.** Correlation matrix heat map (absolute values) of mid-infrared spectra of the laboratory standards in aliphatic C−H region. In this heat map, absolute value of correlation coefficient of absorbances at each wavenumber with absorbances at other wavenumbers is demonstrated (ranging between zero to one).

## Appendix B: Relating mixture property to pure compound property

Laboratory standards which have been used for model development are aerosols of single organic compounds while atmospheric organic aerosols are generally complex mixtures of multitude of species (Hallquist et al., 2009). This fundamental
5  difference highlights the importance of investigating the validity of the models for mixtures. Herein, the validity of the models developed on pure compounds is rationalized mathematically for estimating mean molecular properties of a non-interacting mixture.

In the aliphatic C−H region, a particular absorbance profile is observed due to different absorbance at each wavenumber. The absorbance profile is dependent on areal molar density $n$ (mole per area of the filter) and the absorption coefficient $\varepsilon = \varepsilon(\tilde{\nu})$ of
10  the compound, which is a function of wavenumber ($\tilde{\nu}$). Thus, the absorbance profile $A$ can be written as

$$A = n\varepsilon(\tilde{\nu}), \tag{B1}$$

In this work, spectra are normalized before applying the models. This normalization step is done by a function denoted as $g$. The function $g$ scales the profile between 0 and 1 regardless of the molar abundance, thus is scale-invariance meaning that,

$$g(x) = g(sx), \tag{B2}$$





where $s$ is a an arbitrary scalar. After the normalization step, the model (function) $f$ is applied to the spectra for estimating a molecular property (carbon number or molecular weight) of the laboratory standards or atmospheric samples. $f$ is linear if

$$f\left(\sum_i x_i\right) = \sum_i f(x_i), \tag{B3}$$

which is true for the linear calibration models used in this work. A pure compound $i$ with the absorption coefficient $\varepsilon_i$ is

5 estimated to have the property $\Phi_i$ calculated by a scale-invariant model $f(g(.))$ (combining the model with the normalization step),

$$\Phi_i = f(g(A_i)) = f(g(\varepsilon_i)). \tag{B4}$$

For a mixture, the true mean property $\bar{\Phi}_{true}$ can be written as an molar average of the model estimates for pure compounds assuming no strong interaction between them in the mixture,

$$\bar{\Phi}_{true} = \frac{\sum_i n_i \Phi_i}{\sum_i n_i} = \frac{\sum_i n_i f(g(\varepsilon_i))}{\sum_i n_i} \tag{B5}$$

for which if the model is linear,

$$\frac{\sum_i n_i f(g(\varepsilon_i))}{\sum_i n_i} = f\left(\frac{\sum_i n_i g(\varepsilon_i)}{\sum_i n_i}\right) = \bar{\Phi}_{lin}. \tag{B6}$$

However, when applying the models to a mixture spectrum, the actual value of $\bar{\Phi}$ is estimated from the measured mixture absorbance profile, which is the sum of pure compound spectra, $\sum_i A_i$ as

$$\bar{\Phi}_{mix} = f\left(g\left(\sum_i A_i\right)\right). \tag{B7}$$

Since the normalization function $g$ scales the profile between 0 and 1, i.e. $g(x) = bfx/\max(fx)$, the true mixture mean assuming a linear model will be:

$$\bar{\Phi}_{lin} = f\left(\frac{\sum_i n_i g(\varepsilon_i)}{\sum_i n_i}\right) = f\left(\sum_i \xi_i g(\varepsilon_i)\right) = f\left(\sum_i \frac{\xi_i \varepsilon_i}{\max(\varepsilon_i)}\right), \tag{B8}$$

where $\xi_i = n_i / \sum_i n_i$ is the mole fraction of the $i$th component in the mixture. However, the estimated molecular property for

20 a mixture based on the mixture spectrum ($\bar{\Phi}_{mix}$) is

$$\bar{\Phi}_{mix} = f\left(\sum_i A_i\right) = f\left(\frac{\sum_i n_i \varepsilon_i}{\max(\sum_i n_i \varepsilon_i)}\right) = f\left(\frac{\sum_i \xi_i \varepsilon_i}{\max(\sum_i \xi_i \varepsilon_i)}\right) = f\left(\sum_i \frac{\xi_i \varepsilon_i}{\max(\sum_i \xi_i \varepsilon_i)}\right). \tag{B9}$$

As a result, $\bar{\Phi}_{mix}$ and $\bar{\Phi}_{lin}$ are different because of their different denominators ($\max(\sum_i \xi_i \varepsilon_i)$ and $\max(\varepsilon_i)$). This means that the true mean property of a mixture is not necessarily the property estimated by applying the model to the mixture spectrum. The difference is, however, negligible as long as the models are linear and the compounds in the mixture have relatively similar

25 absorption coefficient. These two conditions are valid for majority of compounds considered in the laboratory standards.

## Appendix C: Elemental carbon and carbon number





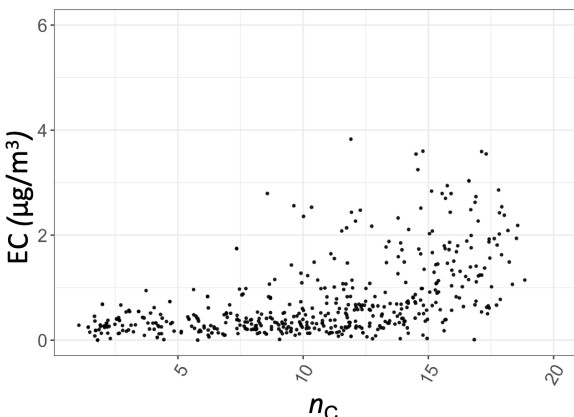

**Figure C1.** Scatter plot showing the relationship between collocated measurements of EC concentration and carbon number estimates by PLS models.

*Author contributions.* AY and ST conceived of the project. AY prepared laboratory standards, performed the calibrations, and analyzed results. AY wrote the manuscript; ST and AMD provided regular input on the analysis and further editing of the manuscript. AMD provided laboratory and ambient sample spectra and ST provided overall supervision of the project.

*Competing interests.* The authors declare no competing interests.

5   *Acknowledgements.* The authors acknowledge funding from the Swiss National Science Foundation (200021_172923) and the IMPROVE program (National Park Service cooperative agreement P11AC91045).



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
