# Peer review of "Estimating mean molecular weight, carbon number, and OM/OC with mid-infrared spectroscopy in organic particulate matter samples from a monitoring network"

_Atmospheric Measurement Techniques, 2020_

## Referee Comment (RC1) · Anonymous Referee #1 · 8 Jun 2020

The paper describes a new method for obtaining important characteristics of Organic Aerosol (OA), such as mean carbon number, molecular weight and organic-mass-to-organic-carbon (OM/OC) ratios, using mid-infrared spectroscopy (also referred to as Fourier transform infrared spectroscopy FTIR). The technique is applicable to spectra acquired non-destructively from Teflon filters used for particulate matter sampling and it is tested on a relevant set of samples (more than 800) coming from the Interagency Monitoring of PROtected Visual Environments (IMPROVE) network in US. The approach involves multivariate statistical analyses (namely Partial Least Squares Re-

gression – PLS) and classification by CART (classification and regression trees) applied on the absorbance profiles and linking them to molecular structures in OM. The multivariate statistical models are trained on calibration spectra prepared from laboratory standards and are then applied to the ambient samples. The results of the models are consistent with previous OM/OC values estimated using different approaches and with temporal and spatial variations in these quantities associated with aging processes, and different source classes (anthropogenic, biogenic, and burning sources).

This is an overall well-written paper even if in some parts (the description of statistical methods for instance) is quite hard to digest and follow and could be improved (look my suggestions below). The method is anyway innovative and informative and so the manuscript is in my opinion worth of publication on AMT after minor changes which are detailed below.

General Comments:

Section 2.4, P9: this section is quite difficult to follow: the steps of the analysis are not clear enough and there are for example some abbreviations not explained (i.e., what does "RMSE" means?) or misleading (i.e., PLS or PLSR?) and some definitions poorly explained. All this makes difficult to follow the statistical methodology, its steps and their meaningfulness. My suggestion is to rephrase the Section, spending time in clarifying the methodology and its steps to make sure the readers can follow your process.

An (incomplete) list of the misleading elements in the section is reported here:

-you define the partial Least Squares Regression as "PLSR", but then you use always "PLS" as abbreviation in the subsequent text. Please decide your favorite abbreviation and check for consistency;

- "RMSE" is not defined;

-readers not familiar with multivariate statistical analysis can find difficult to understand

the concept of "different number of latent variable (LVs)": please explain better what is a latent variable in the context of this analysis and/or the motivation for repeating the analysis with a different number of LVs;

- the unbalanced use of the word "model" (in this section but in general in all the text) makes sometimes difficult to follow the discussion and to understand the different steps of the methodology: the "model" is both the statistical analysis and its results, the calibration process as well as the complete process to determine molecular weight and number of carbon-atoms. The word "model" in the abstract and in the Introduction refers also to thermodynamics and chemical numerical models, making even more confusing the discussion. I suggest to use more carefully the word "model" distinguish between the different types of "models" considered. Other words like "regression" when you are talking of PLS or sometimes simply "analysis" can be used to clarify the steps.

Detailed Comments:

P3, L10: consider to add the article "the" before "spectrum".

P4, L11 & L15: there are question marks inside the brackets: add reference or remove the symbols.

P4, L12: equation (1): consider to move the definition of ïA∎ in a different row.

P6, L5-7: how long is the sampling time? Not clear, even if important to understand possible advantages/disadvantages of the technique. This is especially true because some sentences later (L13 as also shown in Figure 4) it is stated that rural samples have very low recovery. Is this problem possibly fixed by longer sampling time in rural/remote sites? Consider to add 1-2 sentences discussing this here or in the conclusion as a suggestion to make the methodology more robust also in non-urban sites.

Figure 4: what is the number inside the histogram's bars representative for? I suppose it is the number of samples of each category, but this should be described explicitly in the caption.

[Figure]

Section 2.2, P6, l8-: is the choice of the laboratory standards linked to natural abundance of species and/or functional groups? Or what is the rationale in the choice of the laboratory standards? Looking at table 1, why for example only one species of dicarboxylic acid has been tested? Or why only Fructose and not Glucose or Galactose? Or other Sugars with different numbers of C-atoms/molecular weight? I can understand that the choice is made also based on availability of standards and of already existing spectroscopic data, but this should be acknowledged better in the text in my opinion.

P9, L28: "to the classify . . .", please remove "the";

P15, L12: "The is not a concern. . .", not meaningful sentence, probably misspelled;

P16, L2-3: "we used all laboratory standards to produce PLS models to applying to ambient samples", here maybe a passive form is needed. Please, replace with "we used all laboratory standards to produce PLS model to be applied to ambient samples".

P22, L7: other inconclusive question marks. Please replace with the number of figure or explain.
* * *

---

## Referee Comment (RC2) · Anonymous Referee #2 · 19 Jun 2020

Yazdani et al. obtained important characteristics (mean molecular weight, carbon number and OM/OC) of ambient organic particles using the aliphatic C-H absorbance profile in mid-infrared spectrum. The method applied is solid and the analysis is comprehensive with the results clearly presented. The authors also did careful comparison with some previous studies using other techniques. As the molecular weight, carbon number and OM/OC can be used in recent models or parameterizations characterizing organic aerosol (OA) evolution or other physical properties, this study is timely and I recommend the publication after the following comments can be addressed.

Major comments:

It is nice that in the introduction the authors have tried to compare the advantages and disadvantages of several techniques determining organic aerosol compositions, e.g., GC/MS, FT-IR and AMS. However, discussions on soft ionization methods are limited (Line 14-16). In recent years soft ionization methods have been frequently used characterizing elemental compositions of ambient organic aerosols (Mazzoleni et al., 2010; Romonosky et al., 2017) and the elemental composition information has been used predicting physicochemical properties of OA, e.g. volatility (Li et al., 2016; Lin et al., 2016; Xie et al., 2020) and phase state (DeRieux and Li et al., 2018; Li et al., 2020). Though the soft ionization methods have shortcomings such as ionization efficiency as the authors pointed, they give more detailed chemical composition information of OA, i.e., the number of C, H, O, N, S, comparing with the mid-infrared spectroscopy used in the study. I suggest more discussions about the advantages and disadvantages of soft ionization methods and the mid-infrared spectroscopy should be added (Nizkorodov et al., 2011; Laskin et al., 2016). I also suggest the authors could add more discussions about the future development of the mid-infrared spectroscopy, for example, how to characterize the characteristics of nitrogen- and sulfur- containing compounds? The compounds used in this study to produce laboratory standards (Table 1) contain only CH and CHO compounds. Does it mean the method developed in this study can only be applied to CH and CHO compounds? However, ambient OA contain heteroatoms.

Minor comments and technical corrections:

(1) Figure 9: what is the criteria of the liquid and solid phase state? Did the authors measure the viscosity of these compounds or the phase state was estimated? How about the semi-solid phase state, e.g., oil or gel?

(2) Caption of Table 2: better clarify the first 6 principal compounds were listed in Table 1.

(3) Line 13, Page 14: the authors described "Many spectra, particularly urban ones,

are clustered close to tetradecane for the first 4 PCs (Fig. 10)". However, it is difficult to differentiate which points indicate "urban particles" in Fig. 10.

(4) Line 4, Page 11: should be "into" not "in to".

(5) Figure 4 vertical axis: should be "percentage" not "precentage".

(6) Line 13, Page 15: should be "There" not "The is".

(7) Line 4, Page 17: there are two "of" before "the mixture".

References:

DeRieux, W. S. W., Li, Y., Lin, P., Laskin, J., Laskin, A., Bertram, A. K., Nizkorodov, S. A. and Shiraiwa, M.: Predicting the glass transition temperature and viscosity of secondary organic material using molecular composition, Atmos. Chem. Phys., 18, 6331-6351, 10.5194/acp-18-6331-2018, 2018.

Laskin, A., Gilles, M. K., Knopf, D. A., Wang, B. and China, S.: Progress in the analysis of complex atmospheric particles, Annual Review of Analytical Chemistry, 9, 117-143, 2016.

Li, Y., Pöschl, U. and Shiraiwa, M.: Molecular corridors and parameterizations of volatility in the chemical evolution of organic aerosols, Atmos. Chem. Phys., 16, 3327-3344, 10.5194/acp-16-3327-2016, 2016.

Li, Y., Day, D. A., Stark, H., Jimenez, J. and Shiraiwa, M.: Predictions of the glass transition temperature and viscosity of organic aerosols by volatility distributions, Atmos. Chem. Phys. Discuss., 2020, 1-39, 10.5194/acp-2019-1132, 2020.

Lin, P., Aiona, P. K., Li, Y., Shiraiwa, M., Laskin, J., Nizkorodov, S. A. and Laskin, A.: Molecular Characterization of Brown Carbon in Biomass Burning Aerosol Particles, Environmental Science & Technology, 50, 11815-11824, 10.1021/acs.est.6b03024, 2016.

Mazzoleni, L. R., Ehrmann, B. M., Shen, X. H., Marshall, A. G. and Collett, J.

L.: Water-Soluble Atmospheric Organic Matter in Fog: Exact Masses and Chemical Formula Identification by Ultrahigh-Resolution Fourier Transform Ion Cyclotron Resonance Mass Spectrometry, Environmental Science & Technology, 44, 3690-3697, 10.1021/es903409k, 2010.

Nizkorodov, S. A., Laskin, J. and Laskin, A.: Molecular chemistry of organic aerosols through the application of high resolution mass spectrometry, Physical Chemistry Chemical Physics, 13, 3612-3629, 10.1039/c0cp02032j, 2011.

Romonosky, D. E., Li, Y., Shiraiwa, M., Laskin, A., Laskin, J. and Nizkorodov, S. A.: Aqueous Photochemistry of Secondary Organic Aerosol of $\alpha$-Pinene and $\alpha$-Humulene Oxidized with Ozone, Hydroxyl Radical, and Nitrate Radical, The Journal of Physical Chemistry A, 121, 1298-1309, 10.1021/acs.jpca.6b10900, 2017.

Xie, Q., Li, Y., Yue, S., Su, S., Cao, D., Xu, Y., et al. (2020). Increase of high molecular weight organosulfate with intensifying urban air pollution in the megacity Beijing. Journal of Geophysical Research: Atmospheres, 125, e2019JD032200. https://doi.org/10.1029/2019JD032200
* * *

---

## Author Comment (AC1) · 18 Jul 2020

**Response to reviewer comments for manuscript: "Estimating mean molecular weight, carbon number, and OM/OC with mid-infrared spectroscopy in organic particulate matter samples from a monitoring network"**

**Reviewer 1**

The paper describes a new method for obtaining important characteristics of Organic Aerosol (OA), such as mean carbon number, molecular weight and organic-mass-to-organic-carbon (OM/OC) ratios, using mid-infrared spectroscopy (also referred to as Fourier transform infrared spectroscopy FTIR). The technique is applicable to spectra acquired non-destructively from Teflon filters used for particulate matter sampling and it is tested on a relevant set of samples (more than 800) coming from the Interagency Monitoring of PROtected Visual Environments (IMPROVE) network in US. The approach involves multivariate statistical analyses (namely Partial Least Squares Regression – PLS) and classification by CART (classification and regression trees) applied on the absorbance profiles and linking them to molecular structures in OM. The multivariate statistical models are trained on calibration spectra prepared from laboratory standards and are then applied to the ambient samples. The results of the models are consistent with previous OM/OC values estimated using different approaches and with temporal and spatial variations in these quantities associated with aging processes, and different source classes (anthropogenic, biogenic, and burning sources).

This is an overall well-written paper even if in some parts (the description of statistical methods for instance) is quite hard to digest and follow and could be improved. The method is anyway innovative and informative and so the manuscript is in my opinion worth of publication on AMT after minor changes which are detailed below.

We thank the reviewer for the encouraging assessment.

1. Section 2.4, P9: This section is quite difficult to follow: the steps of the analysis are not clear enough and there are for example some abbreviations not explained (i.e., what does "RMSE" mean?) or misleading (i.e., PLS or PLSR?) and some definitions poorly explained. All this makes difficult to follow the statistical methodology, its steps and their meaningfulness. My suggestion is to rephrase the Section, spending time in clarifying the methodology and its steps to make sure the readers can follow your process.

   This section was rewritten with more explanation about each step to make it easier to follow for readers.

2. You define the partial Least Squares Regression as "PLSR", but then you use always "PLS" as abbreviation in the subsequent text. Please decide your favorite abbreviation and check for consistency;

   In the revised version, only "PLSR" has been used to avoid confusion.

3. "RMSE" is not defined;

The definition of "RMSE" was added the to the text. This parameter indicates the root mean square error of predictions of the calibration models that were developed on a part of the calibration set (9/10 of dataset in a 10-fold cross-validation) and used to estimate the desired parameters (molecular weight and carbon number) for the rest of the calibration set (1/10 of dataset in a 10-fold cross-validation).

4. Readers not familiar with multivariate statistical analysis can find difficult to understand the concept of "different number of latent variable (LVs)": please explain better what is a latent variable in the context of this analysis and/or the motivation for repeating the analysis with a different number of LVs;

It was added to the text that LVs are essentially linear combinations of original wavenumbers in the spectra matrix. It is possible to built calibration models using different number of LVs. Models developed using too few LVs do not give a good fit for the calibration set. On the other hand, using too many LVs results in over prediction, i.e good fit for the calibration set (the part of dataset that is used for developing models) but poor predictions for the test set (the part of data set dataset that is not used for model development). As a result, there is an optimum number of LVs, which is identified by a 10-fold cross-validation in this study.

5. The unbalanced use of the word "model" (in this section but in general in all the text) makes sometimes difficult to follow the discussion and to understand the different steps of the methodology: the "model" is both the statistical analysis and its results, the calibration process as well as the complete process to determine molecular weight and number of carbon-atoms. The word "model" in the abstract and in the Introduction refers also to thermodynamics and chemical numerical models, making even more confusing the discussion. I suggest to use more carefully the word "model" distinguish between the different types of "models" considered. Other words like "regression" when you are talking of PLS or sometimes simply "analysis" can be used to clarify the steps.

The word "model" has been used more carefully in the revised version. The statistical models are referred to as "calibration/statistical models" and the term "numerical model" is used whenever referring to numerical simulations to avoid confusion. Models and parametrizations such as 2-D VBS and the carbon number-polarity grid are referred to as "conceptual models".

6. P3, L10: consider to add the article "the" before "spectrum".

Corrected.

7. P4, L11 and L15: There are question marks inside the brackets: add reference or remove the symbols.

This was a Latex compilation error and has been corrected.

8. P4, L12:equation(1): Consider to move the definition of $\mu$ in a different row.

Corrected.

9. P6, L5-7: How long is the sampling time? Not clear, even if important to understand possible advantages/disadvantages of the technique. This is especially true because some sentences later (L13 as also shown in Figure 4) it is stated that rural samples have very low recovery. Is this problem possibly fixed by longer sampling time in rural/remote sites? Consider to add 1-2 sentences discussing this here or in the conclusion as a suggestion to make the methodology more robust also in non-urban sites.

The same protocol was used for the urban and rural sites: samples were collected every third day for 24 hours, midnight to midnight (added to the revised text). Because of lower OM mass

concentration in rural sites the recovery percentage is usually lower. As correctly mentioned by the reviewer, this can be improved by increasing the sampling time at the expense of decreased temporal resolution although monitoring networks are less flexible regarding the protocols (e.g., sampling time). Another problem that remains unresolved even by increasing the sampling time is the low organic-to-inorganic (especially ammonium, which overlaps with the aliphatic $C-H$ absorbances) ratio in those samples that makes the local baseline correction in the 2800–3000 cm$^{-1}$ region complicated due to extensive peak overlap.

10. Figure 4: What is the number inside the histogram's bars representative for? I suppose it is the number of samples of each category, but this should be described explicitly in the caption.

    The numbers represent the number of samples in each category. The information was added to the caption.

11. Section 2.2, P6, L8: Is the choice of the laboratory standards linked to natural abundance of species and/or functional groups? Or what is the rationale in the choice of the laboratory standards? Looking at table 1, why for example only one species of dicarboxylic acid has been tested? Or why only Fructose and not Glucose or Galactose? Or other Sugars with different numbers of C-atoms/molecular weight? I can understand that the choice is made also based on availability of standards and of already existing spectroscopic data, but this should be acknowledged better in the text in my opinion.

    The standards presented in this work include several straight-chain and cyclic alkanes and alkanols combined with the previously existing standards from Ruthenburg et al. (2014). Authors attempted to include standards that are relevant to atmospheric OA (e.g., levoglucosan and sugars which are abundant in biomass burning and alkanes in fossil fuel emission). In addition, it was tried to include a variety of samples necessary for capturing the effects of chain-length, physical phase, cyclic and acyclic structure, and electronegative atoms on the aliphatic $C-H$ profile. However, the availability of standards, their spectroscopic data, and their suitability for atomization were deciding factors for standard selection. This limitation has been acknowledged more clearly in the revised version.

12. P9, L28: "to the classify . . .", please remove "the";

    Corrected.

13. P15, L12: "The is not a concern. . .", not meaningful sentence, probably misspelled;

    "The" was changed to "This".

14. P16, L2-3: "we used all laboratory standards to produce PLS models to applying to ambient samples", here maybe a passive form is needed. Please, replace with "we used all laboratory standards to produce PLS model to be applied to ambient samples".

    The sentence was changed to "all laboratory standards were used to build PLSR models that were applied to the ambient samples".

15. P22, L7: other inconclusive question marks. Please replace with the number of figure or explain.

    This was a Latex compilation error and has been corrected.

**Reviewer 2**

Yazdani et al. obtained important characteristics (mean molecular weight, carbon number and OM/OC) of ambient organic particles using the aliphatic $C-H$ absorbance profile in mid-infrared spectrum. The method applied is solid and the analysis is comprehensive with the results clearly presented. The authors also did careful comparison with some previous studies using other techniques. As the molecular weight, carbon number and OM/OC can be used in recent models or parameterizations characterizing organic aerosol (OA) evolution or other physical properties, this study is timely and I recommend the publication after the following comments can be addressed.

We thank the reviewer for the encouraging assessment.

16. It is nice that in the introduction the authors have tried to compare the advantages and disadvantages of several techniques determining organic aerosol compositions, e.g., GC/MS, FT-IR and AMS. However, discussions on soft ionization methods are limited (Line 14-16). In recent years soft ionization methods have been frequently used characterizing elemental compositions of ambient organic aerosols (Mazzoleni et al., 2010; Romonosky et al., 2017) and the elemental composition information has been used predicting physicochemical properties of OA, e.g. volatility (Li et al., 2016; Lin et al., 2016; Xie et al., 2020) and phase state DeRieux et al. (2018); Li et al. (2020). Though the soft ionization methods have shortcomings such as ionization efficiency as the authors pointed, they give more detailed chemical composition information of OA, i.e., the number of C, H, O, N, S, comparing with the mid-infrared spectroscopy used in the study. I suggest more discussions about the advantages and disadvantages of soft ionization methods and the mid-infrared spectroscopy should be added (Nizkorodov et al., 2011; Laskin et al., 2016).

This is a good point. The comparison was made more complete by mentioning the recent advances and applications of soft ionization methods and also more complete list of advantages and shortcomings of mid-infrared spectroscopy.

17. I also suggest the authors could add more discussions about the future development of the mid-infrared spectroscopy, for example, how to characterize the characteristics of nitrogen- and sulfur- containing compounds? The compounds used in this study to produce laboratory standards (Table 1) contain only CH and CHO compounds. Does it mean the method developed in this study can only be applied to CH and CHO compounds? However, ambient OA contain heteroatoms.

In this study, The mean number of oxygen atoms was estimated indirectly via their effect on the aliphatic $C-H$ absorbances. Authors believe that the method is also applicable to other hetereoatoms. However, the extent of spectral changes in the aliphatic $C-H$ region, is to some extent, dependent on the electronegativity of the heteroatom although some features like peak ratios, which are informative about carbon number, should not be affected by heteroatoms. Since FGs containing other heteroatoms have specific absorbances in mid-infrared spectra (Pavia et al., 2008), the new method might be used in combination with the conventional methods, working based on Beer-Lambert law, to identify heteroatoms (e.g., N in amines, amides, and organonitrates; S in organosulfates) in addition to molecular weight and carbon number. We did not include this discussion in the main text as it is still speculative.

Other interesting and important future aspect of this study is the estimation of OA phase state using spectroscopic features. We found that peak profiles, including peak width, are affected by phase state of the standards. This was added to the text as a future development of the work. In addition a limited analysis regarding phase state estimation using spectroscopic features was added the Supplement (Sect. S3).

18. Figure 9: What is the criteria of the liquid and solid phase state? Did the authors measure the viscosity of these compounds or the phase state was estimated? How about the semi-solid phase state, e.g., oil or gel?

   In this work, the standards (pure compounds) with melting point below the laboratory temperature (25 °C) were considered liquid and vice versa. Compounds such as docosane and docosanol were in the form of amorphous crystals (Arangio et al., 2019) but no semi-solid/viscous compound existed among standards.

19. Caption of Table 2: Better clarify the first 6 principal compounds were listed in Table 1.

   The information was added to the caption.

20. Line 13, Page 14: The authors described "Many spectra, particularly urban ones, are clustered close to tetradecane for the first 4 PCs (Fig. 10)". However, it is difficult to differentiate which points indicate "urban particles" in Fig. 10.

   The points representing urban and rural samples have been color-coded in the revised version.

21. Line 4, Page 11: should be "into" not "in to".

   Corrected.

22. Figure 4 vertical axis: should be "percentage" not "precentage". (6) Line 13, Page 15: should be "There" not "The is".

   Corrected.

23. Line 13, Page 15: should be "There" not "The is".

   Corrected.

24. Line 4, Page 17: there are two "of" before "the mixture".

   Corrected.

**References**

Arangio, A., Delval, C., Ruggeri, G., Dudani, N., Yazdani, A., and Takahama, S.: Electrospray Film Deposition for Solvent-Elimination Infrared Spectroscopy, Appl. Spectrosc., p. 000370281882133, https://doi.org/10.1177/0003702818821330, 2019.

DeRieux, W.-S. W., Li, Y., Lin, P., Laskin, J., Laskin, A., Bertram, A. K., Nizkorodov, S. A., and Shiraiwa, M.: Predicting the Glass Transition Temperature and Viscosity of Secondary Organic Material Using Molecular Composition, Atmos. Chem. Phys., 18, 6331–6351, https://doi.org/10.5194/acp-18-6331-2018, 2018.

Li, Y., Pöschl, U., and Shiraiwa, M.: Molecular Corridors and Parameterizations of Volatility in the Chemical Evolution of Organic Aerosols, Atmos. Chem. Phys., 16, 3327–3344, https://doi.org/10.5194/acp-16-3327-2016, 2016.

Li, Y., Day, D. A., Stark, H., Jimenez, J., and Shiraiwa, M.: Predictions of the Glass Transition Temperature and Viscosity of Organic Aerosols by Volatility Distributions, Atmos. Meas. Tech. Discuss., pp. 1–39, https://doi.org/10.5194/acp-2019-1132, 2020.

Lin, P., Aiona, P. K., Li, Y., Shiraiwa, M., Laskin, J., Nizkorodov, S. A., and Laskin, A.: Molecular Characterization of Brown Carbon in Biomass Burning Aerosol Particles, Environ. Sci. Technol., 50, 11 815–11 824, https://doi.org/10.1021/acs.est.6b03024, 2016.

Mazzoleni, L. R., Ehrmann, B. M., Shen, X., Marshall, A. G., and Collett, J. L.: Water-Soluble Atmospheric Organic Matter in Fog: Exact Masses and Chemical Formula Identification by Ultrahigh-Resolution Fourier Transform Ion Cyclotron Resonance Mass Spectrometry, Environ. Sci. Technol., 44, 3690–3697, https://doi.org/10.1021/es903409k, 2010.

Pavia, D. L., Lampman, G. M., Kriz, G. S., and Vyvyan, J. A.: Introduction to Spectroscopy, Brooks Cole, Belmont, CA, fourth edn., 2008.

Romonosky, D. E., Li, Y., Shiraiwa, M., Laskin, A., Laskin, J., and Nizkorodov, S. A.: Aqueous Photochemistry of Secondary Organic Aerosol of $\alpha$-Pinene and $\alpha$-Humulene Oxidized with Ozone, Hydroxyl Radical, and Nitrate Radical, J. Phys. Chem. A, 121, 1298–1309, https://doi.org/10.1021/acs.jpca.6b10900, 2017.

Ruthenburg, T. C., Perlin, P. C., Liu, V., McDade, C. E., and Dillner, A. M.: Determination of Organic Matter and Organic Matter to Organic Carbon Ratios by Infrared Spectroscopy with Application to Selected Sites in the IMPROVE Network, Atmos. Environ., 86, 47–57, https://doi.org/10.1016/j.atmosenv.2013.12.034, 2014.

Xie, Q., Li, Y., Yue, S., Su, S., Cao, D., Xu, Y., Chen, J., Tong, H., Su, H., Cheng, Y., Zhao, W., Hu, W., Wang, Z., Yang, T., Pan, X., Sun, Y., Wang, Z., Liu, C.-Q., Kawamura, K., Jiang, G., Shiraiwa, M., and Fu, P.: Increase of High Molecular Weight Organosulfate With Intensifying Urban Air Pollution in the Megacity Beijing, J. Geophys. Res. Atmos., 125, e2019JD032 200, https://doi.org/10.1029/2019JD032200, 2020.